# Gender Predilection in Sporadic Parathyroid Adenomas

**DOI:** 10.3390/ijms21082964

**Published:** 2020-04-22

**Authors:** Maria P. Yavropoulou, Athanasios D. Anastasilakis, Argyro Panagiotakou, Evanthia Kassi, Polyzois Makras

**Affiliations:** 1Endocrinology Unit, 1st Department of Propaedeutic Internal Medicine, Medicine School, National and Kapodistrian University of Athens, 11527 Athens, Greece; arpanagiotakou@gmail.com (A.P.); evakassis@gmail.com (E.K.); 2Department of Endocrinology, 424 General Military Hospital, 56429 Thessaloniki, Greece; a.anastasilakis@gmail.com; 3Department of Biological Chemistry, National and Kapodistrian University of Athens, 11527 Athens, Greece; 4Department of Medical Research, 251 Hellenic Air Force & VA General Hospital, 11525 Athens, Greece; pmakras@gmail.com

**Keywords:** sporadic parathyroid adenomas, female, estrogen receptors, progesterone receptors, microRNAs, circular RNAs

## Abstract

Primary hyperparathyroidism is a common endocrinopathy that is mainly caused by benign parathyroid adenomas. The frequency, clinical presentation and complications of the disease show significant differences between genders, with the majority of cases being reported in postmenopausal women. Due to this gender predilection, several studies have investigated the role of sex hormones in the pathogenesis of the disease and their potential use as targets for optimal and gender-specific management. Epigenetic mechanisms that regulate gene transcription may also contribute to these differences between genders. In this review, we outline what is currently known regarding the role of sex hormones and the recent data on the role of non-coding RNAs in the differences between genders in primary hyperparathyroidism due to sporadic parathyroid adenomas.

## 1. Introduction

Primary hyperparathyroidism (pHPT) is one of the most common endocrinopathies in Western countries [1]. It is characterized by excessive secretion of parathyroid hormone (PTH) from the parathyroid glands, leading to hypercalcemia, hypercalciuria and hypophosphatemia. Most cases of pHPT nowadays are asymptomatic, diagnosed incidentally during routine biochemical screening [2] and are caused by the development of benign parathyroid tumors (parathyroid adenomas) (85%) or hyperplasia of the parathyroid glands (15%), with less than 1% being attributed to parathyroid carcinomas with distant metastases [1]. In symptomatic cases, renal and bone manifestations are most commonly seen [3]. The vast majority of cases of pHPT (90–95%) occur sporadically, but they can also occur as part of inherited genetic syndromes caused by mutations in specific target genes, such as in multiple endocrine neoplasia type 1 (MEN 1), type 2A (MEN 2A) and type 4 (MEN 4), familial hypocalciuric hypercalcemia (FHH), neonatal severe hyperparathyroidism (NSHPT), HPT-jaw tumor (HPT-JT) syndrome, and familial isolated pHPT (FIPH) [4]. Compared to sporadic adenomas, pHPT occurring in the context of genetic familial syndromes is mainly caused by multiglandular disease and is characterized by an earlier age of onset [4,5].

Apart from the higher frequency of sporadic cases of pHPT in women, it seems that the clinical presentation and the biochemical profile of the disease also differ between genders [6]. Based on these differences, a potential causality linked to female sex hormones, such as estrogen, especially estradiol (E2) and progesterone (Pg), or their receptors, has been intensively investigated.

In this review, we outline both earlier and current data revisiting the female gender predilection in sporadic parathyroid adenomas in terms of sex hormones and differentially affected epigenetic mechanisms, which may in turn predispose to the differences reported in the frequency and clinical presentation of pHPT between sexes.

## 2. Epidemiology

The prevalence of sporadic pHPT depends on the populations studied and has been reported to range from 1 per 1000 in the United States to 21 per 1000 in Finland [7,8]. Women are affected more frequently than men; however, the ratio between sexes varies in different age groups, from close to equal in subjects younger than 40 years to an almost five-fold higher prevalence in women compared to men in subjects older than 75 years of age [7]. The peak incidence of pHPT is observed among women in the 6^th^ decade of life, when it is about three times more common in women than in men [6,9,10]. In contrast, in hereditary forms of pHPT, both sexes are almost equally affected [11]. The higher prevalence of pHPT of genetic origin at younger ages could contribute to the equal distribution of the disease among sexes in this age group. This hypothesis is further supported by the findings of the US Nationwide Inpatient Sample (NIS), in which the female: male ratio was reported to rise steadily until the perimenopausal age, after which it became stable for the next 20 years before decreasing again [12]. In line with the above, in a Swiss population-based study, the hospitalization rate of patients with pHPT was approximately three times higher in women than in men, peaking in women above the age of 80 [13].

There is considerable discordance among studies regarding the impact of gender on the biochemical profile of pHPT at the time of diagnosis. Circulating PTH has been reported to be similar [6,14], higher [9], or even lower [10] in men compared with women. Serum calcium levels have been found to be similar [6,10,14] or higher [9], serum phosphate levels similar [10] or lower [6], and urinary calcium similar [6] or higher [9,14] in men compared with women.

By contrast, clinical presentation is almost unanimously reported to differ between males and females. Νephrolithiasis appears to be more common in men, while skeletal manifestations are more common in women in most studies [6,9,10], with the exception of one report where men had lower preoperative Z-scores and greater improvement of bone mineral density (BMD) in the femoral neck after parathyroidectomy compared with women [14]. The underlying mechanisms responsible for these differences in clinical presentation among sexes are unclear. However, we should take into consideration that even in eucalcemic individuals, nephrolithiasis is more frequently found in men [15] while osteoporosis occurs more frequently in women [16]. Thus, it might be that the presence of pHPT simply does not affect the gender distribution of these specific complications. There is also discrepancy among studies regarding the role of gender in the frequency of symptoms, with men being reported as either more [6] or less [9] frequently symptomatic than women.

The number of glands involved, and their location, do not seem to be associated with gender [9]. The mean gland weight for a single adenoma was reported to be either higher in males [9] or similar between genders [10]. Surgical outcomes, on the other hand, were equivalent between men and women [9].

## 3. The Role of Female Sex Steroids in the Physiology of Parathyroid Hormone Secretion

Estrogens are a group of steroid hormones that are of critical importance for the reproductive functions of fertile women [17]. Apart from the reproductive organs, which are the main target tissues, estrogens act in a number of other organs and systems, such as the cardiovascular, musculoskeletal and immune systems, the gastrointestinal tract, and the central nervous system [17]. The molecular actions of estrogens are mainly mediated through their nuclear receptors (estrogen receptors; ERs), which are proteins that bind estrogens with high affinity and specificity. Upon binding, ERs act as ligand-modulated nuclear transcription factors that enhance the transcription of target genes [18]. Two ER molecules have been identified, ER-alpha (ERα) and ER-beta (ERβ), which share similar structures with other members of the large family of nuclear receptors. ERβ is further divided into five functional human isoforms, ERβ1, ERβ2, ERβ3, ERβ4 and ERβ5, through alternative spicing [19].

ERβ1 and ΕRα act as DNA-binding transcription factors upon ligand activation. ERβ2 acts as a receptor antagonist for ERα [20]. Molecular modeling has shown that ERβ2 lacks ligand-dependent transcriptional activity, though it does retain some ligand-independent activity [20].

Apart from their well-described classical genomic actions, estrogens can also exert rapid non-genomic actions, which do not involve gene transcription, through rapid induction of the mitogen-activated protein (MAP) kinase and the extracellular signal-regulated kinase (ERK) pathways [21] (Figure 1).

Pg is another endogenous sex steroid involved in the menstrual cycle, pregnancy and embryogenesis in humans. It is mainly responsible for the thickening of the uterus epithelium during the menstrual cycle, where it prepares the endometrial lining to allow implantation of the fertilized ovum. During pregnancy, Pg is produced in the placenta, with its levels remaining elevated throughout pregnancy, where it plays an important role in suppressing the maternal immunologic response to fetal antigens, thereby preventing maternal rejection of the trophoblast [22]. Apart from its essential role in reproduction, it exerts other biological functions in other organ systems, such as mammalian tissue and bone, as well as in the cardiovascular and central nervous systems [23]. Its action is mediated through the nuclear progesterone receptor (PR), which is expressed as two different isoforms, PR alpha (PRα) and PR beta (PRβ), with different molecular weights and distinct functions. PRβ acts as a positive regulator of Pg’s effect, while PRα acts as an antagonist of PRβ [23]. Similarly to estrogens, Pg can also exert rapid non-genomic effects on several tissues through its interaction with cell membrane proteins which induce rapid activation of signal transduction cascades, such as ERK pathways, cAMP/protein kinase A (PKA) signaling, or the phosphatidylinositol 3-kinases (PI3 K)/Akt pathway [23].

Data regarding the role of estrogens, and particularly Ε2 and Pg in PTH secretion from the parathyroid glands in physiological conditions, are to date somewhat controversial.

Early in vitro studies in both animal [24] and human [25] parathyroid cells reported a direct, rapid (within an hour), dose-dependent stimulatory effect of both E2 and Pg on PTH secretion [24,25]. No response of PTH in estrone (E1), estriol (E3), or testosterone was observed, however, and the E2 receptor antagonist tamoxifen did not inhibit the effect of E2 on PTH secretion, leading the authors to hypothesize that the stimulatory effect on PTH might not be exerted through conventional intracellular ERβ signaling [24]. Similar results were reported from in vivo studies with Pg. PRs were identified via in situ hybridization and immunohistochemical assays in the parathyroid glands of female rats [26]. Administration of Pg resulted in a significantly increased mean number of secretory granules in the parathyroid cell cytoplasm in both male and female mice [27], suggesting a direct stimulatory effect of Pg on PTH secretion. These direct effects of E2 and Pg on PTH secretion suggested by early studies were, however, strongly questioned in later in vivo and clinical studies. In a rat model with ovariectomy and hyperparathyroidism induced by chronic kidney disease, exogenous E2 administration significantly decreased PTH mRNA and serum levels indirectly by decreasing the mRNA and serum levels of FGF23 [28].

In an earlier clinical study, significant increases in serum parathyroid PTH concentrations, along with minimal but significant decreases in serum calcium (Ca) levels, were observed in osteoporotic women treated for six months with conjugated equine estrogen [29]; this pointed to an indirect effect of estrogen on PTH secretion, mediated through mild hypocalcemia caused by estrogen-related inhibition of bone resorption [29]. In line with this “indirect effect” hypothesis, in women during the first postmenopausal years, the loss of the restraining effect of estrogen on bone resorption caused an outflow of skeletal calcium into the extracellular fluid, resulting in a compensatory decrease in PTH secretion [30]. No change in PTH levels is also reported in a cross-sectional, population-based study in early postmenopausal women [31].On the other hand, several years post menopause, the loss of the extraskeletal effects of estrogen on intestinal [32] and renal [33] calcium handling leads to a decrease in extracellular calcium concentration and, thus, to a compensatory increase in PTH secretion [31]. In line with the above, transdermal estrogen replacement therapy did not directly affect PTH secretion in postmenopausal women [34] and estrogen treatment or estrogen withdrawal had no significant effect on either basal or ethylenediaminetetraacetic acid-stimulated PTH secretion in elderly postmenopausal women [35]. 

Among healthy fertile women, PTH levels remain stable throughout the menstrual cycle despite the large fluctuation of estrogen levels in most [36,37,38,39,40], but not all [41,42], studies. In one of these studies, an association between PTH and E1, but not E2, was noted [39], while in another, only women presenting with premenstrual syndrome exhibited a mid-cycle elevation of PTH [37]. In contrast, two studies in healthy young women reported a cyclic variation of PTH, peaking around ovulation, which was attributed to the respective fluctuations of E2 levels [41,42].

In pregnancy, a condition of manifold increase in both estrogen and Pg, PTH levels are decreased, although this is probably attributed to the increased production of PTHrP by the placenta [43]. Whether the increased levels of E2 and Pg during pregnancy affect PTH secretion directly or indirectly through the significant increase in PTHrP production currently remains unclear.

## 4. Expression of Female Sex Steroids in Sporadic Parathyroid Tumors

Despite the inconclusive data regarding the role of female hormones in the secretion of PTH from the parathyroid cells under physiological circumstances, the role of E2 in parathyroid tumorigenesis has been extensively investigated in both epidemiological and experimental studies in order to identify gender-specific treatment targets for the management of pHPT.

Most epidemiological studies have reported an inverse relationship between estrogen replacement therapy and the incidence of pHPT [44].

Experimental studies have, however, demonstrated conflicting results. Saxe et al. [45] evaluated 38 abnormal parathyroid tissues (27 adenomas, five with primary hyperplasia and six with secondary hyperplasia) for the presence of ERs and PRs using both the classical hormone-receptor binding assay and an immunohistochemical assay with a monoclonal antibody against nuclear ERs. Only 16% of the samples, mainly from female patients, were ER-positive and 8% were PR-positive. By contrast, Wong et al. [46] demonstrated ERα and ERβ expression in 100% of tissue samples from patients with primary and secondary hyperparathyroidism, although in some cases of secondary hyperparathyroidism the expression was weak. The expression of ERα and two other variants of ERβ, namely ERβ1 and ERβ2, were also studied by Haglund et al. [47] in 37 sporadic parathyroid adenomas at both the mRNA and protein levels. In this study, parathyroid adenomas expressed low levels of the ERα gene, but relatively high levels of the ERβ gene. Similarly, at the protein level, the results from the immunohistochemistry assay showed that all parathyroid adenomas expressed both variants of the ERβ protein, but approximately half of them (43%) expressed the ERα protein. Interestingly, further analysis of ERβ variants revealed that ERβ1 nuclear staining in the tumors was weaker compared to in the normal parathyroid rim (*p* = 0.026), and this difference was more evident in atypical adenomas (*p* = 0.045) and larger tumors (*p* = 0.029) [47]. The results from this study were repeated in a larger number of parathyroid tumors (*n* = 172) and 10 normal parathyroid glands [48]. All normal parathyroid glands and the majority of the parathyroid tumors expressed both ERβ1 and ERβ2 (70.6% expression of ERβ1 and 96.5% expression of ERβ2 in parathyroid tumors) [48]. The expression of both ERβ1 and ΕRβ2 genes, however, was not gender-specific and was not associated with the age or other clinical characteristics of the patients with parathyroid tumors. In addition, ERβ1, but not ΕRβ2, expression was much weaker in parathyroid carcinomas (approximately 50%) and atypical adenomas (approximately 14%) [48]. Similarly to previous results [47], the nuclear expression of ERβ1 was inversely correlated with tumor weight (Spearman’s rank correlation, *r* = −0.20 *p* = 0.011) [48] and was significantly lower in parathyroid tumors compared to in normal parathyroid rim and in normal parathyroid glands [48]. These findings were further supported by an in vitro study in parathyroid tumor cells that demonstrated the coupling of ERβ1 function with tumor suppression [47], lending further support to a linkage between the loss of ERβ1 expression and parathyroid tumorigenesis. The presence of ER gene polymorphisms (namely, PvuII and XbaI) was also investigated in postmenopausal women with pHPT failing, however, to identify significant differences in the distribution of the ER genotypes between pHPT patients and age-matched controls [49].

In general, it seems that parathyroid tumors mainly express ERβ and not ERα. Since ERβ2 cannot mediate estrogen signaling by itself due to its structural properties [50], it appears that the down-regulation of ERβ1 is a mechanism that may effectively silence ligand-mediated estrogen signaling in parathyroid tumors.

## 5. Gender-Based Genetic and Epigenetic Mechanisms in the Pathogenesis of Sporadic Parathyroid Adenomas

The pathophysiology of parathyroid tumorigenesis involves inactivating germline mutations of suppressive oncogenes in the majority of cases with familial predisposition, i.e., in about one-third of sporadic parathyroid adenomas [51] and in approximately 50–75% of parathyroid carcinomas [52]. Genes that have been linked to the pathogenesis of sporadic parathyroid adenomas include *MEN1*, cyclin *D1/PRAD1* and cyclin-dependent kinase inhibitors (*CDKI*), such as *CDKN1B/p27* [4]. However, the genetic role in parathyroid tumorigenesis does not differ between genders, this being in line with what we know about the inherited syndromes, although the number of reported patients is up to now small. Apart from alterations in the *MEN1* gene, genetic abnormalities in other genes appear to occur very rarely. Thus, the role of epigenetic changes in sporadic parathyroid adenomas deserves further investigation.

In recent years, the field of epigenetics, a discipline that seeks, inter alia, to explain significant differences in phenotypes among patients with the same disease, has evolved considerably [53]. Epigenetic mechanisms refer to pathways that influence gene expression in postnatal life without altering the DNA sequence. These mechanisms include DNA methylation, post-translational modifications of histones and post-transcriptional regulation by non-coding RNAs.

In contrast to the other epigenetic mechanisms that modulate gene transcription, non-coding RNAs act at the post-transcriptional level and directly modulate the gene expression of mRNA genes.

Non-coding RNAs, including circular RNAs (circRNAs), long-non-coding RNAs (lncRNAs), and short microRNAs (miRNAs), comprise an invisible layer of signals that control gene expression at various levels in physiology and disease, and possibly in tumorigenesis [54] .

We have previously shown a differential circRNA profile at the tissue level between sporadic parathyroid adenomas and normal parathyroid tissue, suggesting the role of epigenetic changes in the pathogenesis of these tumors [55]. Moreover, we demonstrated a differential expression profile of circRNAs between male and female patients. CircRNAs are formed from the covalent linkage of the 3′ and 5′ ends to form a closed loop [56]. As a result of this closed structure, circRNAs have been shown to be highly stable and largely resistant to RNA degradation pathways [57], which suggests that circRNAs may be more useful molecular biomarkers for human diseases than linear non-coding RNAs. In our study, we identified 19 circRNAs that were significantly up-regulated and four circRNAs that were significantly down-regulated in parathyroid tumors from male patients compared to females [55], but whether these differences may contribute to differences between genders in sporadic parathyroid adenomas remains unknown.

MiRNAs are small, non-coding, single-stranded RNA molecules of 22 nucleotides that bind mRNAs through complementary base pairing, resulting in the suppression of the translation or degradation of mRNAs through RNA-induced silencing complex formation. The differential expression of miRNAs between males and females has been reported, while in various in vitro studies, sex steroid hormones [58], such as E2, Pg and testosterone, have been shown to regulate the expression of specific miRNAs [59,60,61]. Although several studies have identified differential miRNA profiling between normal parathyroid glands and parathyroid tumors, as well as among different types of parathyroid tumors (adenoma vs. carcinoma vs. atypical adenoma) [62,63], the role of miRNAs in mediating gender biases in parathyroid adenoma-induced pHPT is as yet understudied and remains to be elucidated. In our study [55], we also identified several miRNAs that are linked to the profile of differentially expressed circRNAs in male patients compared to female patients with sporadic parathyroid adenomas. Among them, miRNA-1184 was the most frequently reported [55]. The tissue expression of miRNA-1184 has been identified as a key regulator in the progression of prostate cancer by targeting specific transcription factors [64], while it was also found to play a significant role in differentiating high-risk women who develop breast cancer from those who remain cancer-free [65], suggesting that it may be differentially affected by sex hormones.

Both miRNAs and circRNAs have shown gender-associated differences in various tissues and developmental stages, with important biological roles in differentiation, development and aging [26,66], and have been linked to the development of various neoplastic diseases [57].

Taken together, these as yet preliminary data suggest that interactions between circRNAs and miRNAs may contribute to the gender–induced differences in the pathogenesis of various diseases with gender predilection, such as pHPT due to sporadic parathyroid adenomas. Certainly, further research is needed in order to pave the way towards the development of effective gender-based therapeutic strategies.

## 6. Concluding Remarks

Hormonal and genetic differences between males and females have a considerable impact on health and disease. In a number of autoimmune diseases, females exhibit a higher incidence than males [67,68], whereas many types of cancers develop more frequently among males than females [69]. In addition, in several diseases, the incidence also differs according to age, with typical examples being cardiovascular disease, which is more common among men than in premenopausal women [70], and pHPT, which exhibits the opposite gender-specific incidence. In particular, in pHPT, the incidence is higher in postmenopausal women compared to men of a similar age, while both genders seem to be equally affected at younger ages.

The considerable discordance between recent studies demonstrating the expression of ERs in normal parathyroid tissue and parathyroid adenomas [47,48], and earlier studies reporting a lack of ER expression [45,71], is probably attributable to different ER isoforms that were studied. ERβ1, but not ERα, appears to have a role in the pathogenesis and progression of sporadic parathyroid adenomas. Upon ligand activation, ERβ1 translocates to the nucleus, where it acts as a transcription factor. It thus appears that, in cases of parathyroid tumors lacking the expression of ERβ1, the tumor may be unresponsive to the protective and antiproliferative effects of estrogen signaling (Figure 1).

Previous studies of SERMs in pHPT have mainly focused on the pharmacological effects on bone metabolism, serum calcium and PTH secretion, without addressing the potential tumor-suppressive effects of estrogens, which are certainly of equal importance. Evidently, therefore, the potential effect of estrogens on parathyroid tumorigenesis warrants further investigation. In addition, the differential expression of non-coding RNAs, such as micro- and circRNAs, between males and females may be an important underlying mechanism of the gender-biased prevalence and outcome of pHPT.

To sum up, in pHPT, gender differences play a major role in the biochemical and clinical presentation as well as the course of the disease. The exact contribution, however, of the underlying molecular mechanisms, such as the presence of a Y chromosome, an extra X chromosome, a difference in the hormonal milieu, or differences in the epigenetic profile, is still far from being fully elucidated.

Despite recent advances in molecular biology and genetics, personalized and gender-specific treatment for diseases with a clear role of gender in their pathogenesis and outcome is still far from being implemented in everyday clinical practice and, therefore, remains an area requiring future research.

## Figures and Tables

**Figure 1 ijms-21-02964-f001:**
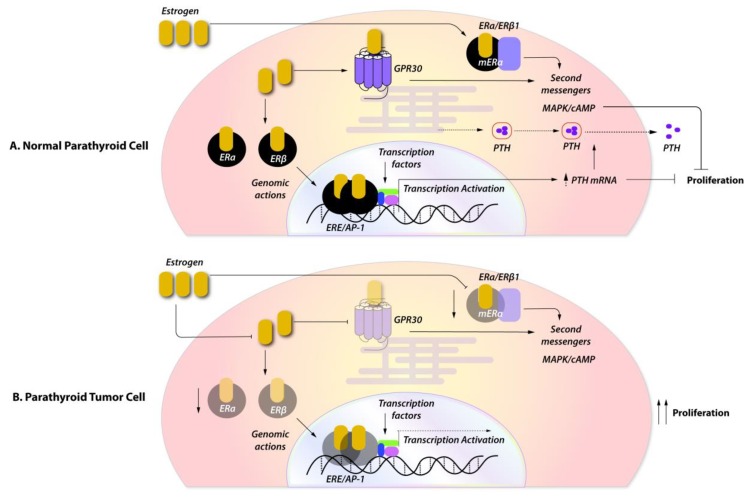
The probable role of estrogens in the pathogenesis of parathyroid adenomas. (**A**) In normal parathyroid cells, estrogen signaling through ERα and ERβ1 exerts its genomic and non-genomic effects by increasing PTH secretion and release, and controlling proliferation. (**B**) In parathyroid tumors, the low expression of ERβ1 and the absence of ERα attenuates the protective antiproliferative effect of estrogens and enhances tumor growth. ERα and ERβ1, estrogen receptor alpha and beta 1; GPR30, G protein-coupled receptor for estrogens; ERE/AP1, estrogen responsive element/activating protein 1; mERα and β1, membrane estrogen receptor alpha and beta 1; MAPK, mitogen-activated protein kinase; cAMP, cyclic adenosine monophosphate.

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
