# Peer review of "Gender Predilection in Sporadic Parathyroid Adenomas"

_ijms, 2020, doi:10.3390/ijms21082964_

Round 1

Reviewer 1 Report

The authors review an interesting topic, the gender predilection for sporadic parathyroid adenomas. They describe the role of female sex hormones as well as the possible role of epigenetic factors.  

It seems a bit coincidental that the authors chose to describe both sex hormones as well as epigenetic changes as possible explanation for the gender differences. Can they explain why they chose this? They should also clarify this in the text.

Introduction line 48:

…they can also occur’ . With this phrase it seems that parathyroid adenomas are seen in disease like MEN, although this consists of polyglandular hyperplasia. Please rephrase.

Epidemiology:

The fact that genetic disease is more common in parathyroid disease at younger age may explain the observation that gender differences are less obvious at younger age. Can the authors comment on this? This is also supported by reference number 13. It would be logic to mention this first, and give reference 12 also as support (otherwise I do not see the importance of this reference in here).

It would be interesting if the authors could speculate on the underlying mechanisms that could explain the observed differences between gender with respect to the clinical presentation (line 82-92).  

Line 106: is it important to describe all isoforms of the ER? The authors do not come back to this point.

There are various paragraphs on ER, without a clear subject for each paragraph, and no specific goal that reflects the importance of the information for this review. E.g. line 102/103 and line 108 state the same. This entire part needs to be rephrased.

Line 120: Progesteron is involved in the menstrual cycle, pregnancy etc. But what is the effect? If the authors describe the ligand/receptor of progesterone and estrogen in such detail, it would be nice if the authors could explain what the general effects are.

The authors describe the possible effect of E2 (which should be introduced as estradiol within the text) and Pg on PTH secretion from line 133 to link 168. The paragraph is difficult to understand. It would be nice if the authors could describe their interpretation of the data more clearly, together with the implications, and rephrase the text.  

Line 165: The authors describe that during pregnancy, estrogen and Pg are increased and PTH levels are decreased. What is the importance of this with respect to the parathyroid gland and E2/Pg? It is well known that PTHrP is responsible for the decrease in PTH, which needs to be added to the text.

Expression of female sex steroids on sporadic parathyroid tumors.

The first part of this topic, divided in various paragraphs (which actually appear as one paragraph rather than 3 different paragraphs), is again difficult to read.

Concluding remarks: this part is very nicely written.

Figure 1. the Figure is very nice and really adds to the understanding of the text.

Author Response

Reviewer 1

The authors review an interesting topic, the gender predilection for sporadic parathyroid adenomas. They describe the role of female sex hormones as well as the possible role of epigenetic factors.

  1. It seems a bit coincidental that the authors chose to describe both sex hormones as well as epigenetic changes as possible explanation for the gender differences. Can they explain why they chose this? They should also clarify this in the text.

Response:

We thank the reviewer for this comment. The idea of including epigenetic changes as part of the gender differences in pHPT stemmed from our previous study, where we identified significant differences in the expression profile of circular non-coding RNAs in parathyroid tumors (tissue samples) between males and females. The data are scarce in this field. However, although the hypothesis that epigenetic changes may be involved in the gender predilection of sporadic parathyroid adenomas is still in its infancy, we wanted to present this idea, as it is a novel research path and we think it is highly interesting. We have now revised the respective section in order for the reader to follow more easily our approach to this matter (pages 8-9, lines 245-279).

  1. Introduction line 48:

…they can also occur’. With this phrase it seems that parathyroid adenomas are seen in disease like MEN, although this consists of polyglandular hyperplasia. Please rephrase.

Response: We thank the reviewer for this comment. We have rephrased the paragraph accordingly in order to be more accurate regarding the cases of pHPT found in familial syndromes, which are mainly due to parathyroid hyperplasia (page 2, lines 44-54).

  1. Epidemiology:

The fact that genetic disease is more common in parathyroid disease at younger age may explain the observation that gender differences are less obvious at younger age. Can the authors comment on this? This is also supported by reference number 13. It would be logic to mention this first, and give reference 12 also as support (otherwise I do not see the importance of this reference in here).

Response:

Following the reviewer’s suggestion, we have accordingly rephrased the text in the respective part of the revised manuscript (pages 2-3, lines 72-78).

  1. It would be interesting if the authors could speculate on the underlying mechanisms that could explain the observed differences between gender with respect to the clinical presentation (line 82-92).

Response:

We have now added some lines in the revised text speculating on the possible underlying mechanisms in response to this reviewer’s suggestion (page 3 lines 89-93).

5.Line 106: is it important to describe all isoforms of the ER? The authors do not come back to this point.

Response:

We describe at this point all isoforms of ER because, as we point out later, it concerns the two isoforms of ERβ (1 and 2) that demonstrated significant expression in parathyroid adenomas and not ERα, as originally suspected and investigated.

  1. There are various paragraphs on ER, without a clear subject for each paragraph, and no specific goal that reflects the importance of the information for this review. E.g. line 102/103 and line 108 state the same. This entire part needs to be rephrased.

Response:

We have now rephrased this part in order to focus better on the information we need to provide to enable the reader to more clearly understand the particular findings of ER in parathyroid adenomas (pages 3 and 4, lines 102-119).

  1. Line 120: Progesterone is involved in the menstrual cycle, pregnancy etc. But what is the effect? If the authors describe the ligand/receptor of progesterone and estrogen in such detail, it would be nice if the authors could explain what the general effects are.

Response:

We have now rephrased and expanded the relative paragraph in order to include some general information on the role of progesterone in the menstrual cycle and pregnancy (page 4, lines 120-125).

  1. The authors describe the possible effect of E2 (which should be introduced as estradiol within the text) and Pg on PTH secretion from line 133 to link 168. The paragraph is difficult to understand. It would be nice if the authors could describe their interpretation of the data more clearly, together with the implications, and rephrase the text.

Response:

We have now introduced E2 as estradiol with its first appearance in the text (Introduction, page 2, line 58) and reconstructed the paragraph according to your suggestion. We first report data from in vitro studies suggesting a direct effect of E2, Pg on PTH (1st paragraph), then in vivo data (2nd paragraph), and finally clinical data questioning the direct effect from studies correlating E2 levels with PTH levels (3rd paragraph) and from clinical interventional studies evaluating the effect of E2 administration on PTH (4th paragraph). We hope it is clearer now.

We did not mark in yellow color the changes in the respective section as there are many and this could result in yellow marking of a large part of the revised manuscript (pages 4-5, lines 137-172).

9.Line 165: The authors describe that during pregnancy, estrogen and Pg are increased and PTH levels are decreased. What is the importance of this with respect to the parathyroid gland and E2/Pg? It is well known that PTHrP is responsible for the decrease in PTH, which needs to be added to the text.

Response:

We thank the reviewer for this comment. We have included the role of PTHrP in PTH suppression during pregnancy (page 6, lines 174-177).

10.Expression of female sex steroids on sporadic parathyroid tumors.

The first part of this topic, divided in various paragraphs (which actually appear as one paragraph rather than 3 different paragraphs), is again difficult to read.

Response:

We have now revised this section in order to make it more concrete and easier to read (page 6-7, lines 193-217).

  1. Concluding remarks: this part is very nicely written.

Response:

We thank the reviewer for her/his kind comment.

  1. Figure 1. the Figure is very nice and really adds to the understanding of the text.

Response:

Again, we thank the reviewer for her/his kind comment.

Reviewer 2 Report

The Authors thoroughly reviewed a niche but important topic. They evaluated literature data concerning the potential role of sex hormones and epigenetic mechanisms that could explain the prevalence among women of sporadic parathyroid adenomas.

The review is narrative in nature and includes all the currently available key papers on the topic. The results obtained are interesting.
Only minor language editing is needed.

Author Response

Reviewer 2

The Authors thoroughly reviewed a niche but important topic. They evaluated literature data concerning the potential role of sex hormones and epigenetic mechanisms that could explain the prevalence among women of sporadic parathyroid adenomas.

The review is narrative in nature and includes all the currently available key papers on the topic. The results obtained are interesting.

We thank the reviewer for her/his kind remarks.

Only minor language editing is needed.

Response:

The revised form of our manuscript has now been reviewed by a professional native English speaker.